# Inferring the Future by Imagining the Past

**Kartik Chandra***
MIT CSAIL

**Tony Chen***
MIT Brain and Cognitive Sciences

**Tzu-Mao Li**
UC San Diego

**Jonathan Ragan-Kelley**
MIT CSAIL

**Joshua Tenenbaum**
MIT Brain and Cognitive Sciences

## Abstract

A single panel of a comic book can say a lot: it can depict not only where the characters currently are, but also their motions, their motivations, their emotions, and what they might do next. More generally, humans routinely infer complex sequences of past and future events from a *static snapshot of a dynamic scene*, even in situations they have never seen before.

In this paper, we model how humans make such rapid and flexible inferences. Building on a long line of work in cognitive science, we offer a Monte Carlo algorithm whose inferences correlate well with human intuitions in a wide variety of domains, while only using a small, cognitively-plausible number of samples. Our key technical insight is a surprising connection between our inference problem and Monte Carlo path tracing, which allows us to apply decades of ideas from the computer graphics community to this seemingly-unrelated theory of mind task.

## 1 Introduction

Hemingway's shortest short story simply reads "For sale: baby shoes, never worn." There is no action in this sentence—it is just a description of a static scene—but as readers, we nonetheless infer a complex and tragic backstory from the "snapshot" that Hemingway describes. This remarkable ability comes naturally to humans: we routinely reconstruct motives from evidence (e.g. at a crime scene), recognize intentions from unfinished tasks (e.g. grading incomplete homework), and enjoy artistic depictions of dynamic action in static media (e.g. a Renaissance "tableau" or sculpture).

How do we do it, so quickly and so intuitively? A long line of empirical work in cognitive science has shown that people, even infants and young children [Southgate and Csibra, 2009, Gergely and Csibra, 2003, Jara-Ettinger et al., 2015, Hamlin et al., 2007], make rapid, flexible, and robust theory-of-mind judgments "out of the box": in novel domains they have never seen before, and without extensive pre-training on data. This remarkable ability has motivated decades of work in AI (see Section 4), which has successfully addressed the problem of inferring an agent's goal from a *trajectory* of observed actions in a few- or even one-shot manner [Baker et al., 2017, 2009]. These methods cast the problem of action understanding as Bayesian inference: $P(\text{goal} \mid \text{actions}) \propto P(\text{actions} \mid \text{goal})P(\text{goal})$, where $P(\text{actions} \mid \text{goal})$ is modeled by comparing the observed actions to the optimal actions a rational agent would take towards that goal.

However, if we only observe a single state snapshot, these methods break down—there are simply no actions to condition on. Instead, we must jointly infer not only where the agent might be going, but also where it came from. Recently, Lopez-Brau et al. [2020, 2022] performed this inference by rejection-sampling possible paths taken by the agent. Their model's predictions are remarkably close to human judgements. However, rejection sampling is extremely inefficient—it is slow even on simple problems, and simply does not scale to more sophisticated problems, suggesting that there is more to how humans perform such inference.

37th Conference on Neural Information Processing Systems (NeurIPS 2023).

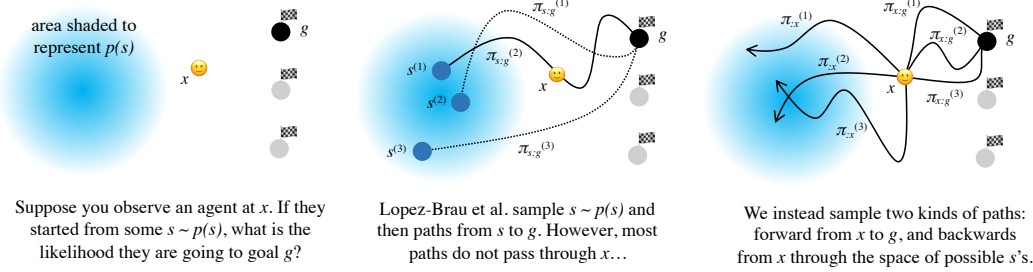

Suppose you observe an agent at $x$. If they started from some $s \sim p(s)$, what is the likelihood they are going to goal $g$?

Lopez-Brau et al. sample $s \sim p(s)$ and then paths from $s$ to $g$. However, most paths do not pass through $x$…

We instead sample two kinds of paths: forward from $x$ to $g$, and backwards from $x$ through the space of possible $s$'s.

Figure 1: How can we infer what an agent is trying to do, based on a snapshot of its current state?

In this paper, we propose a dramatically more efficient algorithm that recovers human-like judgements even on significantly more sophisticated problems. We take inspiration from the field of computer graphics, which has developed a wealth of Monte Carlo algorithms for "path tracing." **Our key insight is an analogy between tracing paths of *light* through 3D scenes and paths of *agents* through planning domains.** This insight allows us to import decades of algorithmic advances from computer graphics to the seemingly-unrelated social intelligence problem of snapshot inference.

In Section 2 we formalize this problem and present a new Monte Carlo algorithm for sampling approximate solutions. Then, in Section 3, we show that our algorithm is up to $30,000\times$ more efficient than prior work. Finally, in Section 3.3 we demonstrate via three behavioral studies that our model's predictions match human judgements on new, scaled-up tasks inaccessible to prior work.

## 2 Our proposed algorithm

We describe our method in the setting of Markov Decision Processes (MDPs). An MDP is a tuple $\langle \mathcal{S}, \mathcal{A}, \mathcal{T}, \mathcal{R}, \gamma \rangle$, where $\mathcal{S}$ is the set of possible states, $\mathcal{A}$ is the space of possible actions, $\mathcal{T} : \mathcal{S} \times \mathcal{A} \times \mathcal{S} \to [0,1]$ gives the transition probability, $\mathcal{R} : \mathcal{S} \times \mathcal{A} \to \mathbb{R}$ is the reward function and $\gamma \in [0,1]$ is a discount factor. We will specifically consider MDPs that have a single "goal state" $g$ with positive reward, such as a grid world with a target square the agent needs to reach. We will assume that $g$ is known to the agent, but not to us (the observers).

Consider an agent that begins in some initial state $s \sim p(s)$. While the agent is on its trajectory from $s$ to $g$, we observe a "snapshot" of the agent in some interim state $x$. Given only $x$ (and not $s$!), our aim is to infer $p(g \mid x)$.

Applying Bayes' rule, we have $p(g \mid x) \propto p(x \mid g)p(g)$, where $p(g)$ is our prior belief about what $g$ might be. The likelihood $p(x \mid g)$ can be computed by marginalizing over all possible start states $s$ and all possible paths $\pi_{s:g}$ from $s$ to $g$, where a path $\pi_{s:g}$ is defined as a sequence of states $\pi_{s:g} = (s_1, \ldots s_n)$ such that $s_1 = s$ and $s_n = g$. Performing this marginalization, and noting that $p(s \mid g) = p(s)$ because $s$ and $g$ are independent, we have:

$$p(x \mid g) = \int_s p(x \mid s, g)p(s \mid g)ds = \int_s \int_{\pi_{s:g}} \overbrace{p(x \mid \pi_{s:g}, s, g)}^{\text{likelihood of snapshot}} \underbrace{p(\pi_{s:g} \mid s, g)}_{\text{likelihood of path}} p(s)d\pi_{s:g}ds \quad (1)$$

To evaluate the likelihood of a snapshot $p(x \mid \pi_{s:g}, s, g)$, we apply the *size principle* [Tenenbaum, 1998, 1999, Griffiths and Tenenbaum, 2006], analogous to the *generic viewpoint assumption* in computer vision [Freeman, 1994, Albert and Hoffman, 2000]. In this case, the principle states that the snapshot was equally likely to have been taken anywhere along the path, and therefore the likelihood of a snapshot conditioned on a path is inversely proportional to the length of the path. If $\delta(x \in \pi)$ indicates whether path $\pi$ passes through $x$, and $|\pi|$ indicates the length of $\pi$, then $p(x \mid \pi_{s:g}, s, g)$ is given by $\delta(x \in \pi_{s:g})|\pi_{s:g}|^{-1}$. As a consequence of this definition, all paths that do not pass through $x$ contribute zero probability, and thus can be dropped from the integral.

To evaluate the likelihood of a path $p(\pi_{s:g} \mid s, g)$, we apply the *principle of rational action:* agents are likelier to take actions that maximize their utility [Dennett, 1989, Jara-Ettinger et al., 2016]. There are many ways to formalize this intuition. A common option, which we adopt, is to say that at each step, the agent chooses an action with probability proportional to the softmax over its $Q$-values at its current state, with some temperature $\beta$. That is, $p(x \to x' \mid g) \propto \sum_a \exp(\beta Q_g(x, a))\mathcal{T}(x \to x' \mid a)$, where $\mathcal{T}(x \to x' \mid a)$ is the transition probability from $x$ to $x'$ if action $a$ is taken, and $p(\pi \mid g) \propto \Pi_t p(x_t \to x_{t+1} \mid g)$.

We now have all the ingredients we need to evaluate $p(x \mid g)$. However, to compute it exactly we would need to integrate over all possible initial states $s$, and the set of paths $\pi_{s:g}$, which could be infinite (agents might wander for arbitrarily long, albeit with vanishingly low probability). To approximate the likelihood in finite time, Lopez-Brau et al. turn to Monte Carlo sampling (Algorithm 1). They rejection-sample paths $\pi_{s:g}$ by sampling a candidate start state $s^{(i)} \sim p(s)$, simulating a rollout of the agent to sample a path $\pi_{s:g}^{(i)} \sim p(\pi_{s:g}^{(i)} \mid s^{(i)}, g)$, and then averaging the integrand over these samples. With $N$ samples, their unbiased likelihood estimator is given by $\hat{p}(x \mid g) = \frac{1}{N} \sum_{i=1}^{N} \delta(x \in \pi_{s:g}^{(i)})|\pi_{s:g}^{(i)}|^{-1}$.

Unfortunately, in practice this scheme is extremely slow: even in a $7 \times 7$ gridworld with fewer than 49 states (only 2 of which were possible initial states), Lopez-Brau et al. report taking over 300,000 trajectory samples per goal to perform inference. In the rest of this section, we will describe a series of algorithmic enhancements that allow for comparable inference quality with just 10 samples per goal (i.e. 30,000$\times$ fewer). We will develop our algorithm (Algorithm 2) through three insights.

## 2.1 First insight: only sample paths through the observed state

Our first insight is that $\delta(x \in \pi)$ is extremely sparse—most paths likely do not pass through $x$, and so most naïve path samples contribute zero to the estimator. We would like to only sample paths that pass through $x$. Any such path can be partitioned at $x$ into two portions, $\pi_{s:x}$ and $\pi_{x:g}$. Let us integrate separately over those portions.

$$p(x \mid g) = \int_s \int_{\pi_{s:x}} \int_{\pi_{x:g}} \frac{p(\pi_{s:x} \mid g) \, p(\pi_{x:g} \mid g)}{|\pi_{s:x}| + |\pi_{x:g}|} p(s) \, d\pi_{x:g} \, d\pi_{s:x} \, ds \qquad (2)$$

This suggests a more efficient Monte Carlo sampling scheme. Rather than rejection-sampling paths $\pi_{s:g}^{(i)}$ directly from $s$ to $g$, we can independently sample two path segments: a "past" path $\pi_{s:x}^{(i)}$ from $s$ to $x$, and a "future" path $\pi_{x:g}^{(i)}$ from $x$ to $g$. Any such path is guaranteed to pass through $x$, so no samples are wasted.

However, we now have two new problems. First, it is not clear how to sample paths $\pi_{s:x}^{(i)}$ from $s$ to $x$, because rollouts of a simulated agent are unlikely to pass through $x$ on their way to $g$. We could imagine using a *second* planner just to chart paths from $s$ to $x$, but this would require a lot of additional planning work. Second, we still have to sample $s$. If the space of initial states is small (e.g. a room only has one or two doors), then this is no issue. However, in practice this space might be very large or even infinite. For example, if you observe someone driving to work in the morning, their home could be anywhere in the city. Furthermore, most of these states might be inaccessible or otherwise implausible, and it would be a waste of computational resources to consider them. In the next section, we show how to solve both of these problems by tracing paths *backwards in time*.

## 2.2 Second insight: sample $\pi_{:x}$ backwards in time

Our second insight is that we can collapse the first two integrals by jointly integrating over the domain of all paths $\pi_{:x}$ that terminate at $x$, no matter where they started from. Say a path $\pi_{:x}$ begins at $\pi_{:x}[0]$. Then, we can rewrite our likelihood as below.

$$p(x \mid g) = \int_{\pi_{:x}} \int_{\pi_{x:g}} \frac{p(\pi_{:x} \mid g) \, p(\pi_{x:g} \mid g)}{|\pi_{:x}| + |\pi_{x:g}|} p(\pi_{:x}[0]) \, d\pi_{x:g} \, d\pi_{:x} \qquad (3)$$

This suggests that we should sample $\pi_{:x}^{(i)}$ *backwards* through time, starting from $x$. No matter how we extend this path, we obtain a valid path from $\pi_{:x}^{(i)}[0]$ to $x$. We illustrate this in Figure 1.

An analogy to path tracing in computer graphics is helpful here. When rendering a 3D scene, a renderer must integrate over all paths of light that begin at a light source in the scene and end at a pixel on the camera's film—a problem formalized by the rendering equation [Kajiya, 1986]. Of course, these paths may be reflected and refracted stochastically by several surface interactions along the way. Rather than starting at one of the millions of light sources in the scene and tracing a ray hoping to eventually reach the camera film, renderers instead start at the camera and trace rays *backward into the scene* until they inevitably reach a light source.

Similarly, here we trace paths backwards from $x$ into the past—$s$ corresponds to a light source, each action taken by the agent corresponds to a stochastic surface interaction, and $x$ corresponds to a pixel on the camera. Indeed, our integral is analogous to the rendering equation, bringing to our disposal the entire Monte Carlo light transport toolbox—a toolbox the rendering community has spent decades developing. (These techniques were pioneered by Veach [1998], though see Pharr et al. [2016] for an accessible review.) The particular ideas we borrow are the following:

**Importance sampling**   The first idea is to be deliberate about how paths are extended backwards in time. We could sample predecessor states uniformly at random—however, that would lead to "unlikely" or irrational paths. Instead, we preferentially select a predecessor state $x_{\text{prev}}$ based on the likelihood of transitioning to the current state from $x_{\text{prev}}$. Then, we re-weight our path sample appropriately so that our estimator remains unbiased (see Algorithm 2, line 14).

**Russian roulette termination**   The next concern is when to stop extending a path into the past. In principle, paths could be infinitely long in some domains. However, at some point paths become so unlikely that extending them is not worth the computational effort of tracing them. An unbiased finite-time solution to this problem is given by the *Russian roulette* method [Carter and Cashwell, 1975, Arvo and Kirk, 1990]: at each step, we toss a weighted coin, and only continue extending the path if it comes up "heads." Then, we weight subsequent samples appropriately to keep the estimator unbiased (see Algorithm 2, line 11).

**Bidirectional path tracing**   The last concern is that the path may not ever "find" the region of state space where the agent could have started. Consider a setting where the space of possible initial states is large but sparse—for example, if we know the agent started from a *red* house somewhere in the city. Importance sampling path predecessors with Russian roulette termination is not guaranteed to stop at a red house, whereas forward-sampling paths from a random red house is not guaranteed to pass through the observed state. In computer graphics, this situation is analogous to rendering a scene with many lights that are occluded from the camera. The classic solution is *bi-directional path tracing* [Lafortune and Willems, 1993, Veach and Guibas, 1995]: first, we "cache" some paths forward-simulated from randomly sampled lights (see Algorithm 3). Then, when backwards-tracing rays, we find opportunities to connect them to a continuation in the cache (see Algorithm 2, line 8).

As a correctness check, we compare the numerical results from our method and the rejection sampling approach of Lopez-Brau et al. [2020] in Appendix B, showing that likelihoods computed from the two methods match closely as expected.

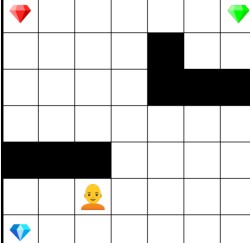 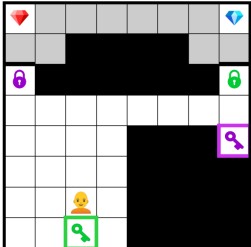

Figure 2: **(left)** In this example of the "grid" domain, we observe an agent near the blue gem. Even though we do not know where the agent started from, our intuition says that the agent is heading towards the blue gem. **(right)** In this example of the "keys" domain, we observe an agent right next to the green key. Humans infer that the agent is heading towards the green key because it wants the blue gem. Our algorithm replicates both of these inferences with only 10 samples.

**Algorithm 1** Rejection sampling, as in prior work. Compare to our proposed method, Algorithm 2.

**Input:** $x$, the agent's current state (e.g. position in gridworld)
$\quad\quad\quad$ $g$, the hypothesized goal
$\quad\quad\quad$ $P(s \to s' \mid g)$, the probability the agent will move to $s'$ from $s$
$\quad\quad\quad$ $P_{\text{start}}(s)$, the prior over the agent starting at $s$
**Output:** $\widehat{p}(x|g)$
1: $t \leftarrow 0, n \leftarrow 0$, sample $x_{\text{current}}$ with probability $\propto P_{\text{start}}(\cdot)$
2: **while** $x_{\text{current}}$ is not an end state **do**
3: $\quad$ **if** $x_{\text{current}} = x$ **then**
4: $\quad\quad$ $n \leftarrow n + 1$
5: $\quad$ sample $x_{\text{next}}$ with probability $p_{\text{choice}} \propto P(x_{\text{current}} \to \cdot \mid g)$
6: $\quad$ $x_{\text{current}} \leftarrow x_{\text{next}}$ and $t \leftarrow t + 1$
7: **return** $1 \,/\, \ell$ if $n > 0$, otherwise $0$

---

**Algorithm 2** Our bidirectional likelihood sampler

**Input:** $x$, $g$, $P(s \to s' \mid g)$, $P_{\text{start}}(s)$ as in Algorithm 1
$\quad\quad\quad$ $\alpha$, the strength of importance sampling
$\quad\quad\quad$ $d$, an average termination depth for Russian roulette
$\quad\quad\quad$ $C$, an optional bidirectional path tracing cache (see Algorithm 3)
**Output:** $\widehat{p}(x|g)$
1: $\ell \leftarrow 0$
2: $t_{\text{next}} \leftarrow 0, x_{\text{current}} \leftarrow x$ $\quad\quad\quad\quad\quad\quad\quad\quad\quad\quad\quad\quad$ ▷ *Sample forward from $x$ to $g$*
3: **while** $x_{\text{current}}$ is not an end state **do**
4: $\quad$ sample $x_{\text{next}}$ with probability $p_{\text{choice}} \propto P(x_{\text{current}} \to \cdot \mid g)$
5: $\quad$ $x_{\text{current}} \leftarrow x_{\text{next}}$ and $t_{\text{next}} \leftarrow t_{\text{next}} + 1$
6: $t_{\text{prev}} \leftarrow 1, x_{\text{current}} \leftarrow x, p_\pi \leftarrow 1$ $\quad\quad\quad\quad\quad\quad\quad\quad$ ▷ *Sample backwards from $x$*
7: **while** true **do**
8: $\quad$ **if** $x_{\text{current}} \in C$ **then** $\quad\quad\quad\quad\quad\quad$ ▷ *Check BDPT cache for available completions*
9: $\quad\quad$ sample $(t_{\text{cache}}, w)$ from $C[x_{\text{current}}]$
10: $\quad\quad$ **return** $w \cdot (\#C[x_{\text{current}}]/\#C) \cdot p_\pi/(t_{\text{cache}} + t_{\text{prev}} + t_{\text{next}})$
11: $\quad$ **if** flip$() < 1/d$ **then** $\quad\quad\quad\quad\quad\quad\quad\quad$ ▷ *Russian roulette termination*
12: $\quad\quad$ **return** $P_{\text{start}}(x_{\text{current}}) \cdot p_\pi/(t_{\text{prev}} + t_{\text{next}}) \cdot 1/(1/d)$ $\quad$ ▷ *Record sample starting at $x_{\text{current}}$*
13: $\quad$ $p_\pi \leftarrow p_\pi \,/\, (1 - 1/d)$ $\quad\quad\quad\quad\quad\quad\quad\quad\quad$ ▷ *Apply Russian roulette weight*
14: $\quad$ sample $x_{\text{prev}}$ with probability $p_{\text{choice}} \propto \exp(\alpha \cdot P(\cdot \to x_{\text{current}} \mid g))$ $\quad$ ▷ *Choose predecessor*
15: $\quad$ $p_\pi \leftarrow p_\pi \cdot P(x_{\text{prev}} \to x_{\text{current}} \mid g) \,/\, p_{\text{choice}}$ $\quad\quad\quad$ ▷ *Apply importance sample weight*
16: $\quad$ $x_{\text{current}} \leftarrow x_{\text{prev}}$ and $t_{\text{prev}} \leftarrow t_{\text{prev}} + 1$
17: **return** $\ell$

---

## 2.3 Third insight: this algorithm allows for on-line planning

Our method is agnostic to the algorithm used to compute the agent's policy, which allows us to choose from any combination of reinforcement-learning or heuristic search approaches. Model-based methods like value iteration require an expensive pre-computation of the value function for all possible goals and states, even though some of those states may not ever be visited by our algorithm. It seems implausible that humans do this, because we make judgements quickly even in new domains. Moreover, adding obviously-irrelevant states should not make the problem harder to solve—we should simply not bother planning from those states until they somehow become relevant.

Motivated by this, in domains that can be expressed as classical planning problems (i.e. domains with a single "goal state" with positive reward), we use an online A* planner instead of precomputing policies by value iteration. We define $p(x \to x' \mid g)$ to be a softmax over the difference in path costs between $x$ and $x'$ to $g$ as computed by the planner: $p(x \to x' \mid g) \propto \sum_a \exp\left(\beta(C(x \to g) - C(x' \to g))\right)$. That is, the agent is likelier to move to states that will bring it closer to the goal. To avoid re-planning from scratch for every evaluation of $p(x_{t-1} \to x_t \mid g)$, we run A-star *backwards* from the goal to the current state. This lets us re-use the bulk of its intermediate computations (known distances, evaluations of the heuristic, etc.) between queries.

---

**Algorithm 3** Grow the bidirectional path tracer's cache (to be called repeatedly)

---

**Input:** $g$, $P(s \rightarrow s' \mid g)$, $P_{\text{start}}(s)$ as in Algorithm 1, $d$ as in Algorithm 2, and $C$, a cache
1:   $t \leftarrow 0$, $w \leftarrow 1$, sample $x_{\text{current}}$ with probability $\propto P_{\text{start}}(\cdot)$
2:   **while** $x_{\text{current}}$ is not an end state **do**
3:      add $(t, d \cdot w)$ to $C[x_{\text{current}}]$
4:      sample $x_{\text{next}}$ with probability $\propto P(x_{\text{current}} \rightarrow \cdot \mid g)$
5:      **if** flip() $< 1/d$ **then**
6:        **break**
7:      $w \leftarrow w \, / \, (1 - 1/d)$
8:      $x_{\text{current}} \leftarrow x_{\text{next}}$ and $t \leftarrow t + 1$

---

## 3   Experiments

To evaluate our sampling algorithm, we chose a suite of benchmark domains reflecting the variety of inferences humans make:

**Simple gridworld**    We re-implement the $7 \times 7$ gridworld domain from Lopez-Brau et al. (Figure 2). The agent seeks one of three gems and at every timestep can move north, south, east or west. The viewer's inference task is to determine which gem the agent seeks. While Lopez-Brau et al. fix two possible starting-points ("entryways") for the agent, our method can optionally relax this constraint and instead have a uniform prior over the start state.

**Doors, keys, and gems (multi-stage planning)**    This is a more advanced $8 \times 8$ gridworld, inspired by Zhi-Xuan et al. [2020]. The agent is blocked from its gem by *doors*, which can only be opened if the agent is has the correct *keys* (the agent always starts out empty-handed). The inference task is to look at a snapshot image and determine which gem the agent seeks. For example, if we observe the snapshot in Figure 2, we might infer that the agent plans to get the green key to obtain the blue gem.

**Word blocks (non-spatial)**    In this domain, the agent spells a word out of the six letter blocks by picking and placing them in stacks on a table. However, they are interrupted (e.g. by a fire alarm) and have to leave the room before finishing. The inference tasks are to look at the blocks left behind and determine (a) which word the agent was trying to spell, and (b) which blocks the agent has touched.

**Additional domains**    Appendix C shows results from additional domains from the cognitive science literature that further reflect the flexibility of our method. We demonstrate domains with partial observability (C.1), multiple agents (C.2), and continuous state spaces with physics (C.3). The breadth of our experiments reflects the typical scope of related work in cognitive science.

### 3.1   Qualitative analysis

Tables 1 and 2 show some example inferences made by our algorithm. Each cell is colored according to the posterior distribution over goals. If the inference algorithm produced no valid samples for a cell, that cell is marked with an $\times$ symbol. With just 10 samples, our method's posterior inferences are near-convergent and align well with human responses. In comparison, with 10 samples rejection sampling typically produces extremely noisy predictions, and often simply fails to produce any non-rejected samples at all—indeed, in the blocks domain even 1,000 samples are not always enough to produce a single non-rejected sample.

### 3.2   Quantitative analysis

We report the total variation $\text{TV}(x) = \frac{1}{2} \sum_{g_i} |\hat{p}(g_i \mid x) - p(g_i \mid x)|$ between the true posterior and inferences made using 10 samples of both our method and rejection sampling, averaged for 100 trials and across all of the inference tasks in the benchmark. We take the true posterior to be our method's estimate with 1,000 samples, though we also show results if the true posterior is taken to be rejection sampling with 10,000 samples. Our results are shown in Table 3. Across all domains, our algorithm substantially outperforms rejection sampling.

Table 1: Qualitative comparison of inference algorithms. Cells are colored based on the posterior distribution over goals if the agent is observed in that cell. Cells marked × had all samples rejected. Gray shaded cells were excluded from analysis.[2] We show results for 10 samples and 1,000 (1k) samples, comparing rejection sampling as described by Lopez-Brau et al. [2020], our method, and human subjects. **We produce near-convergent human-like inferences with only 10 samples.**

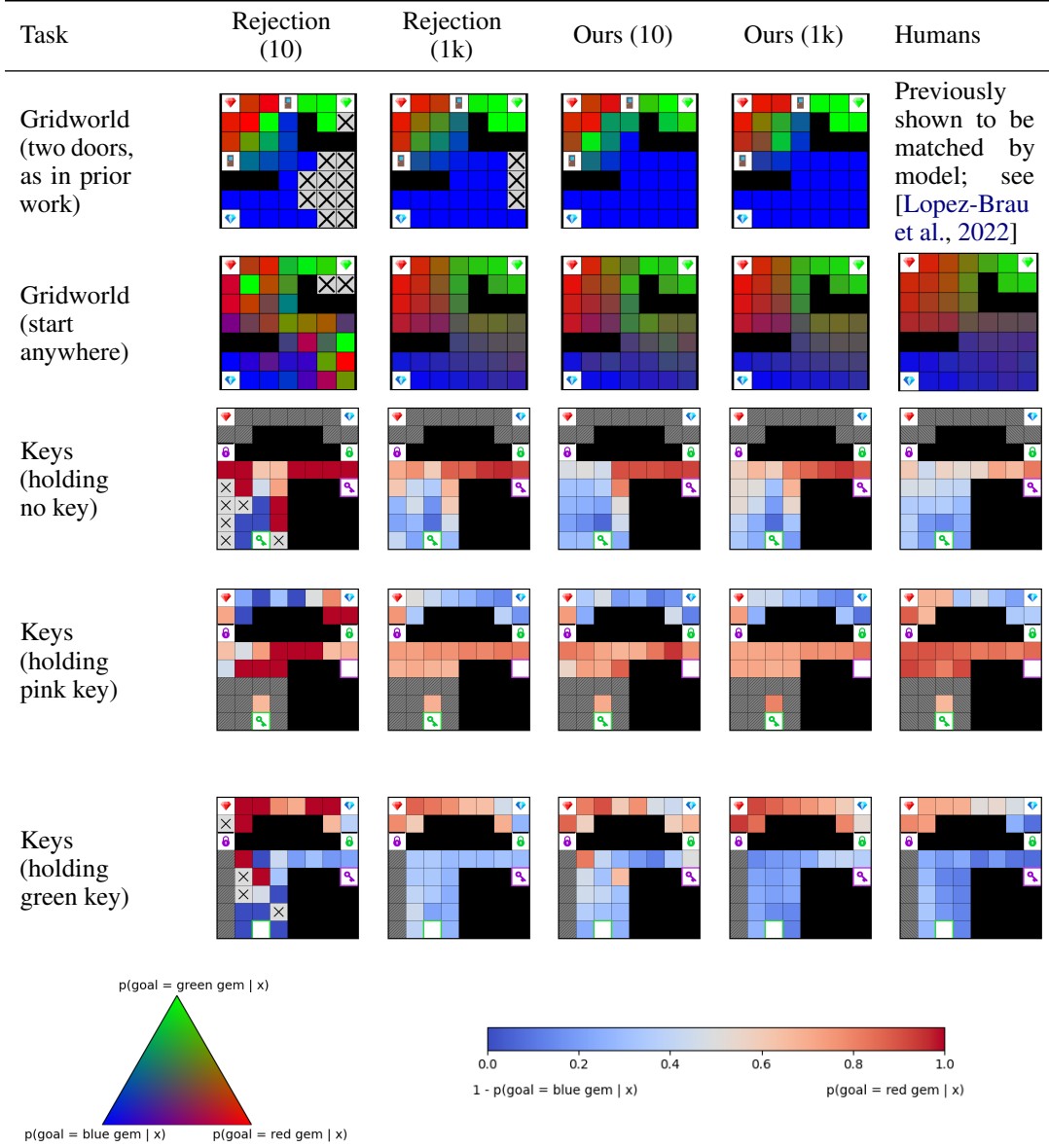

| Task | Rejection (10) | Rejection (1k) | Ours (10) | Ours (1k) | Humans |
|---|---|---|---|---|---|
| Gridworld (two doors, as in prior work) | | | | | Previously shown to be matched by model; see [Lopez-Brau et al., 2022] |
| Gridworld (start anywhere) | | | | | |
| Keys (holding no key) | | | | | |
| Keys (holding pink key) | | | | | |
| Keys (holding green key) | | | | | |

p(goal = green gem | x)

p(goal = blue gem | x)      p(goal = red gem | x)

0.0   0.2   0.4   0.6   0.8   1.0

1 - p(goal = blue gem | x)      p(goal = red gem | x)

## 3.3 Comparison to human judgements

We recruited $N = 200$ participants and collected judgements for a variety of "snapshots" in each of our domains (see experimental design in Appendix A). We found that our model predicts human intuitions well (see last columns of Tables 1 and 2), with strong correlations across domains (see Figure 3). Thus, our work not only replicates the findings of previous work [Lopez-Brau et al., 2020, 2022], but also shows that those findings continue to hold in domains that previous inference algorithms could not scale to.

---

[2]Shaded cells in Row 3 are excluded because they are inaccessible without holding a key. Shaded cells in Row 4 are excluded because if the agent already picked up the pink key earlier, there is no reason to then move

Table 2: Qualitative comparison of inference algorithms. Blocks are colored according to inferred probability of that block having been touched by the person stacking the blocks (red is high probability, blue is low). We show results for 10 samples and 1,000 (1k) samples, comparing rejection sampling, our method, and human subjects. **We produce near-convergent inferences with only 10 samples.** In comparison, rejection sampling is unable to make any inference with 10 samples, and sometimes even fails with 1,000 samples. When it succeeds, its predictions are high-variance and overconfident.

| Rejection (10) | Rejection (1k) | Ours (10) | Ours (1k) | Humans |
|---|---|---|---|---|

# 4 Related work

**Cognitive science** Human social cognition and "theory of mind" are well-modeled by **Bayesian inverse planning** [Baker et al., 2009, Jara-Ettinger, 2019, Baker et al., 2017], which infers an agent's goals from its observed actions. Inverse planning predicts human inferences about multiple agents [Kleiman-Weiner et al., 2016, Wu et al., 2021], social scenes [Ullman et al., 2009, Netanyahu et al., 2021], emotion [Ong et al., 2015], and agents who make mistakes [Zhi-Xuan et al., 2020].

A growing body of work studies how people make inferences about the past from a "snapshot" of present physical evidence [Smith and Vul, 2014, Gerstenberg et al., 2021, Lopez-Brau and Jara-Ettinger, 2020, Paulun et al., 2015, Yildirim et al., 2017, 2019]. Recently, Lopez-Brau et al. [2020,

---

far away from the doors. Similarly, in Row 5, there is no reason to move that far left after picking up the green key. We found that querying repeatedly at those extreme locations confuses human participants about the task setup. The single cell above the green key in Row 4 is an exception we wished to test: one might think from that position that the agent is trying to collect both keys, leading to uncertainty about the goal. But people (and the model) agree that the agent still wants the red gem.

Table 3: Quantitative comparison of inference algorithms (see Section 3.2). We show the average total variation distance (TV) of a 10-sample posterior estimate against a converged posterior, averaged over 100 trials. **Lower is better.** Our method does significantly better than rejection sampling [Lopez-Brau et al., 2022] for each task, whether the converged "ground truth" is computed with 1,000 samples using our method, or 10,000 samples using rejection sampling.

| Benchmark (TV against Ours @ 1k) | Rejection @ 10 | Ours @ 10 |
|---|---|---|
| Grid (two doors) | 0.063 | 0.0257 |
| Grid (starting anywhere) | 0.159 | 0.0538 |
| Keys (observed holding no key) | 0.409 | 0.108 |
| Keys (observed holding pink key) | 0.388 | 0.157 |
| Keys (observed holding green key) | 0.381 | 0.119 |
| Blocks | 0.985 | 0.358 |
| Benchmark (TV against Rejection @ 10k) | Rejection @ 10 | Ours @ 10 |
| Grid (two doors) | 0.0705 | 0.0336 |
| Grid (starting anywhere) | 0.153 | 0.0836 |
| Keys (observed holding no key) | 0.672 | 0.410 |
| Keys (observed holding pink key) | 0.456 | 0.382 |
| Keys (observed holding green key) | 0.463 | 0.312 |
| Blocks | *Rejection @ 10k did not converge* | |

2022] asked how people make inferences about the past and future of agents from static physical evidence they leave behind, which is something even children can do [Pelz et al., 2020, Jacobs et al., 2021]. We build on this line of work by dramatically accelerating inference, and extending it to more sophisticated domains where previous methods could not scale.

**Artificial intelligence**  Bayesian approaches have been successful in approaches to **plan recognition**, the problem of inferring an agent's plan from observed actions [Ramírez and Geffner, 2009, 2010, Sohrabi et al., 2016, Charniak and Goldman, 1993]. Our work provides a method for plan recognition from a single state snapshot, with no need to observe actions.

The reinforcement learning community has long sought to learn an agent's reward function by observing the actions it takes via **inverse reinforcement learning** or IRL [Ng et al., 2000, Arora and Doshi, 2021, Ziebart et al., 2008]. Recently, Shah et al. [2019] proposed "IRL from a single state." Their method, "Reinforcement Learning by Simulating the Past" (RLSP), learns reward functions based on a single observation of a human in the environment, assuming that the observation is taken at the end of a finite-horizon MDP of fixed horizon $T$. We build on RLSP in three ways: **(1)** RLSP is highly sensitive to the time horizon hyperparameter $T$. We dispense with the fixed-horizon assumption altogether, integrating over past trajectories of all possible lengths. **(2)** Unlike Shah et al., we do not

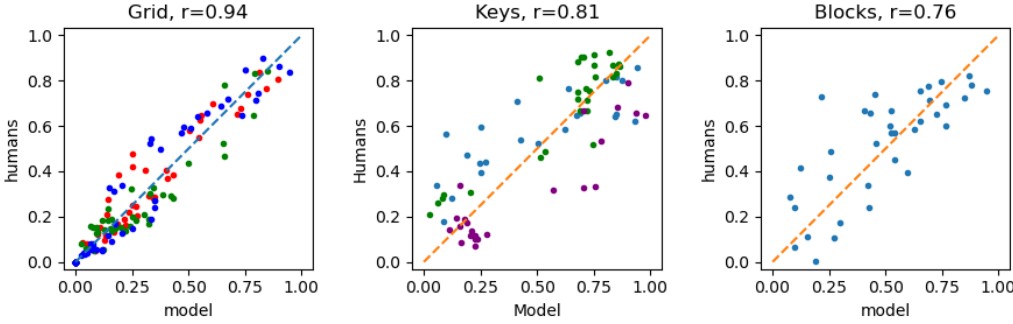

Figure 3: Our inferences correlate well with human responses. In the "grid" plot, colors represent goal gem colors. In the "keys" plot, colors represent which keys (if any) the agent is seen holding, and we show $p(\text{goal is blue gem} \mid x)$. In the "blocks" plot, we show $p(\text{touched} \mid x)$ for each block.

assume the snapshot was taken at the end of the agent's journey—the agent can be observed partway. **(3)** Our sampling-based method scales to significantly larger domains, because we do not have to integrate exhaustively over all possible trajectories.

The Deep RLSP algorithm of Lindner et al. [2020] also addresses point (3) above by sampling past trajectories. However, while Deep RLSP's maximum-likelihood formulation is designed to recover robot policies, our Bayesian approach is designed specifically to model human intuitions about agents' goals. Thus, unlike Deep RLSP, our method incorporates priors over start states in a principled way, and produces human-like uncertainty quantification.

## 5 Limitations and future work

**Sampling over goals**    In this paper, we showed how to scale inference to a large set of possible initial states. However, we compute the posterior by enumeration over all possible goals, which takes linear time in the size of the space of goals. To scale to larger goal spaces as well, it may be possible to use Markov Chain Monte Carlo to sample goals conditioned on the observed state, similar to what Veach and Guibas [1997] propose for path tracing in computer graphics. This may seem challenging because we only stochastically estimate the likelihood $p(x \mid g)$. However, because our estimator is unbiased, it is still possible to use it to compute valid Metropolis-Hastings transitions [Andrieu and Roberts, 2009].

**Cognitive plausibility**    Our method's sample efficiency suggests that it may resemble how humans actually do this task, similar to how few-shot algorithms have shown promise in modeling human behavior in other domains [Vul et al., 2014]. Following previous work using eye-tracking studies to investigate cognitive processes underlying simulation and planning [Gerstenberg et al., 2017], we hope to use eye-tracking to compare human strategies to our algorithm. In the language of Marr [1982], this would allow us to go beyond the *computational* account of Lopez-Brau et al. [2020] and take a first step towards an *algorithmic* account of inverse planning with snapshot states.

**Theoretical analysis**    The *time complexity* of taking a single sample using our algorithm is straight-forward: it scales linearly with the Russian roulette depth and the number of possible goals (although sub-linear runtime in number of goals is possible through sampling as described above). However, the more pertinent question is that of *sample complexity*. While our algorithm is unbiased, analyzing the rate of convergence as a function of the size of the sample space and complexity of the domain are topics for future work.

**Investigating potential applications**    While our goal in this paper was to model how humans perform this inference task, extensions of our method could be applied to a variety of practical problems: some examples include understanding dynamic action in photographs of sports, interpreting step-by-step instruction manuals, and forensic reconstruction based on available static evidence.

**From analysis to synthesis**    We began this paper with Hemingway's short story—and indeed, more generally, artists have long depicted dynamic action through static scenes [McCloud, 1993]. In future work we hope to consider the inverse problem of designing interesting, evocative, or ambiguous scenes by optimizing *over* inference [Chandra et al., 2023a,b, 2022], and designing expressive and human-interpretable robotic "gestures" similar to legible planning [Dragan et al., 2013].

## 6 Conclusion

In this paper, we offered a cognitively-plausible algorithm for making inferences about the past and the future based on the observed present. Building on prior work from both the cognitive science and AI communities, and drawing inspiration from the Monte Carlo rendering literature in computer graphics, we presented a highly sample-efficient method for performing such inferences. Finally, we showed that our method matched human intuitions on a variety of challenging inference tasks to which previous methods could not scale.

## Acknowledgments and Disclosure of Funding

We thank Joe Kwon, Tan Zhi-Xuan, Alicia Chen, Max Siegel, Lionel Wong, and John Cherian for thoughtful conversations as we developed these ideas. This research was funded by NSF grants #CCF-1231216, #CCF-1723445 and #2238839, and ONR grant #00010803. Additionally, KC was supported by the Hertz Foundation, the Paul and Daisy Soros Fellowship, and an NSF Graduate Research Fellowship under grant #1745302, and TC was supported by an NDSEG Fellowship.

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

## A  Experimental design

For each of our experiments reported in Section 3.3, we recruited $N = 200$ participants from Prolific [Palan and Schitter, 2018]. Participants were paid $15 per hour ($1.25 total for blocks and grid domains, and $2.00 for the keys domain), and our experiments were conducted with IRB approval.

Participants were first familiarized with the environment, through both text instructions and a sample video of an agent performing the task in the domain. Then, they were told that their objective was to infer the agent's goal from a single snapshot. They answered several questions to check their comprehension of both the domain and the task they were asked to perform, and were not allowed to continue unless they answered the comprehension questions correctly. **The full experimental design is available in HTML format in the supplementary materials.** No data was excluded from our analyses.

## B  Numerical test of correctness

Programming sophisticated importance sampling routines is a challenging and bug-prone engineering effort [Cusumano-Towner et al., 2019, Anderson et al., 2017, Pharr et al., 2016]. To test that our algorithm is unbiased, i.e. that it produces correct likelihoods in expectation, we compared likelihoods computed by rejection sampling and our sampler using converged estimates (25,000 samples each). For this experiment we used a uniform $4 \times 4$ grid-world, with the prior on start states being uniform along the first row ($x = 0$) and the goal being the far corner $(3, 3)$. The results of this experiment are shown in Figure 4. Our estimator has a dramatically different implementation than rejection sampling (compare Algorithms 1 and 2). However, the computed likelihoods are indistinguishable at every cell in the grid, even in "corner-case" cells such as the goal cell itself. **This provides a strong check that our algorithm and its implementation are both indeed correct.**

## C  Additional domains

We used our algorithm to perform inferences in three additional domains. The purpose of these domains is to show the remarkable flexibility of our method: how it can make interesting inferences in a wide variety of settings. Though we did not collect human subject data for these domains, we show results for cases where the inference task is relatively straightforward.

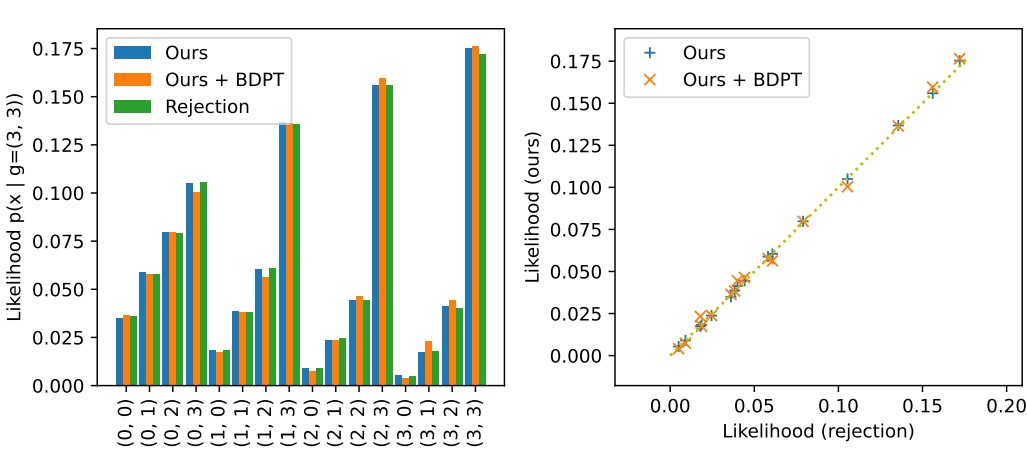

Figure 4: Our sampler's likelihoods precisely match rejection sampling, with and without bidirectional path tracing, giving a strong numerical check of our method's correctness (Appendix B).

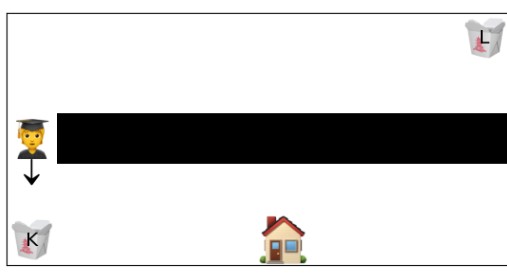 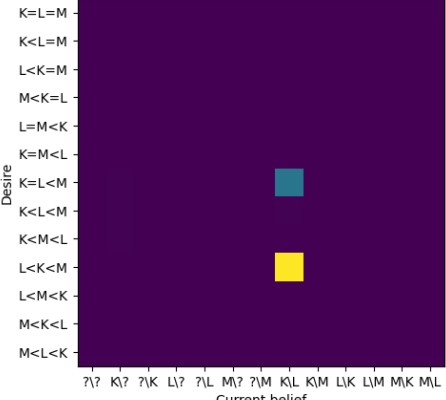

Figure 5: The student is observed heading south around the wall. A rational inference is that the student started at home, and went around the wall to check what the far food truck was. Seeing that it was Lebanese and not Mexican (their favorite), the student disappointedly turns around to make peace with the nearby Korean food. **As shown on the heatmap to the right, our model captures this joint belief-desire inference, predicting that the student now knows what is at both trucks, and reconstructing the student's likely preference ordering over the three cuisines.** *Note: the belief label "K \ ?" means that the student thinks the south-west parking spot has a Korean food truck parked, but is unsure about the north-east parking spot.* See Appendix C.1.

### C.1 Food trucks (joint belief/desire inference)

The food trucks domain, taken from the cognitive science literature [Baker et al., 2017], is a Partially Observable Markov Decision Process (POMDP). It consists of a $5 \times 10$ gridworld with an opaque wall in the middle. A hungry graduate student wakes up at home (one side of the wall) and wishes to eat at a food truck. There are two parking spots where food trucks usually park, and three kinds of food trucks that could be parked at each of those spots: Korean, Lebanese, and Mexican (K, L, and M). The graduate student might have preferences among the cuisines, but might also be uncertain about which trucks are parked at each spot today. Thus, they might engage in information-seeking behavior by looking behind the opaque wall, and then choosing a food truck to walk to based on their preferences. **The inference task is to determine (a) the student's preferences over food trucks, and (b) the student's (current) belief state about which truck is at each parking spot.**

Using this domain, Baker et al.'s inverse planning model was able to jointly infer the student's beliefs and desires from an observed trajectory; those inferences closely matched responses from human subjects. Here, we perform the same type of inference, but from a single observed snapshot.

For example, in the example in Figure 5, the student is observed moving south next to the wall. A Korean food truck is parked in the southwest parking spot, and a Lebanese food truck is parked in the northeast spot. Seeing this scene, a reasonable inference is that the student went looking around the wall to see if the Mexican food truck (their favorite) was parked on the other side. Seeing that it was Lebanese food instead, the student turns around and makes peace with the nearby Korean food. Indeed, our model captures this inference: in the joint posterior distribution over both beliefs and desires, our model is confident that the student now knows that the northeast truck has Lebanese food, and furthermore that the student's favorite food is Mexican.

A more sophisticated inference emerges if the student is observed moving *north* instead of south (Figure 6). Now, a reasonable inference is that the student dislikes Korean food, and is going around the wall to check what is at the other truck. The model captures this: it favors the hypothesis that the student is unsure what is at the northeast truck, and also places high weight on Korean being the least favorite food option.

However, as is visible on the right half of the heatmap, the model also places some weight on the possibility that the student knows that there is Lebanese food and prefers it, or that the student (mistakenly) believes there is Mexican food and prefers that.

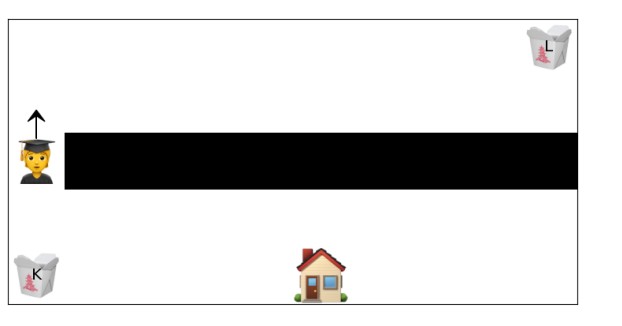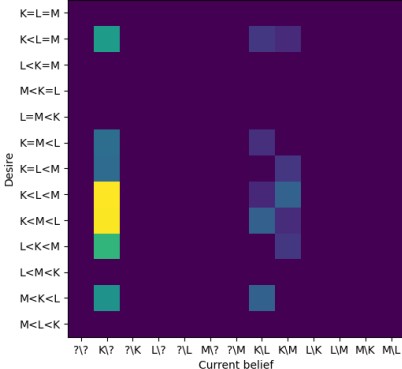

Figure 6: Here, the student is observed going north instead of south. A more sophisticated inference emerges, showing that the student is likely uncertain about which truck is parked behind the wall. See Appendix C.1

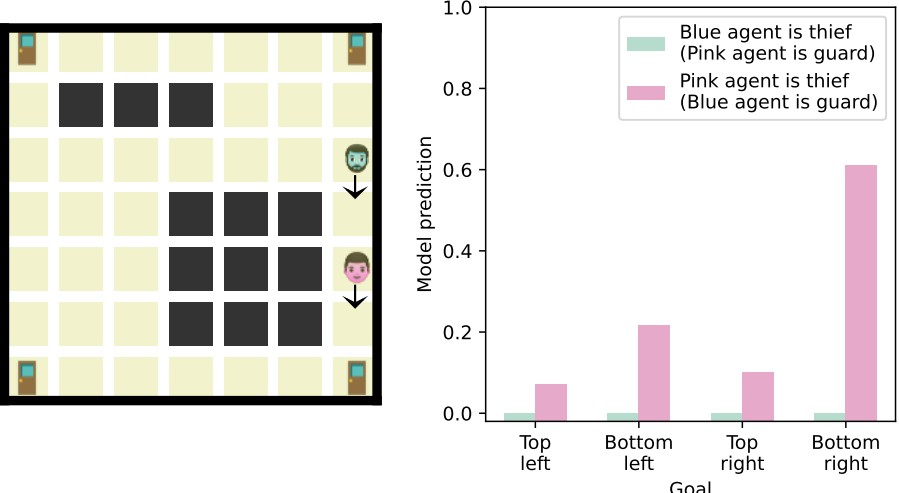

Figure 7: Two agents are observed by a security camera in an art museum. Who is the guard, who is the thief, and where is the thief trying to escape to? **Our model predicts that the guard is the blue agent, the thief is the pink agent, and that the exit is in the bottom right.** See Appendix C.2.

## C.2 Heist (multi-agent domain)

In this multi-agent domain inspired by classic stimuli in cognitive science [Baker et al., 2008, Southgate and Csibra, 2009, Heider and Simmel, 1944], two agents—blue and pink—occupy a $7 \times 7$ gridworld representing an art museum. One of the agents is a "thief," whose objective is to escape the museum by reaching the exit, and the other is a "guard," whose objective is to catch the thief. There are four doors in the room, only one of which is an exit, and the rest of which are dead ends. Both agents know which door is the exit, but this information is *not* visible to the observer (all doors are rendered identically). **The inference tasks are to look at a snapshot of the two agents and jointly infer (a) which agent is the thief and which is the guard, and (b) which door is the exit.**

In the example in Figure 7, it is clear from the snapshot that the blue agent is the guard and is chasing the pink agent, the thief, to the bottom-right corner. The model reproduces this inference, though also acknowledges the possibility that the thief might actually be heading onward past the bottom-right, to the bottom-left corner instead.

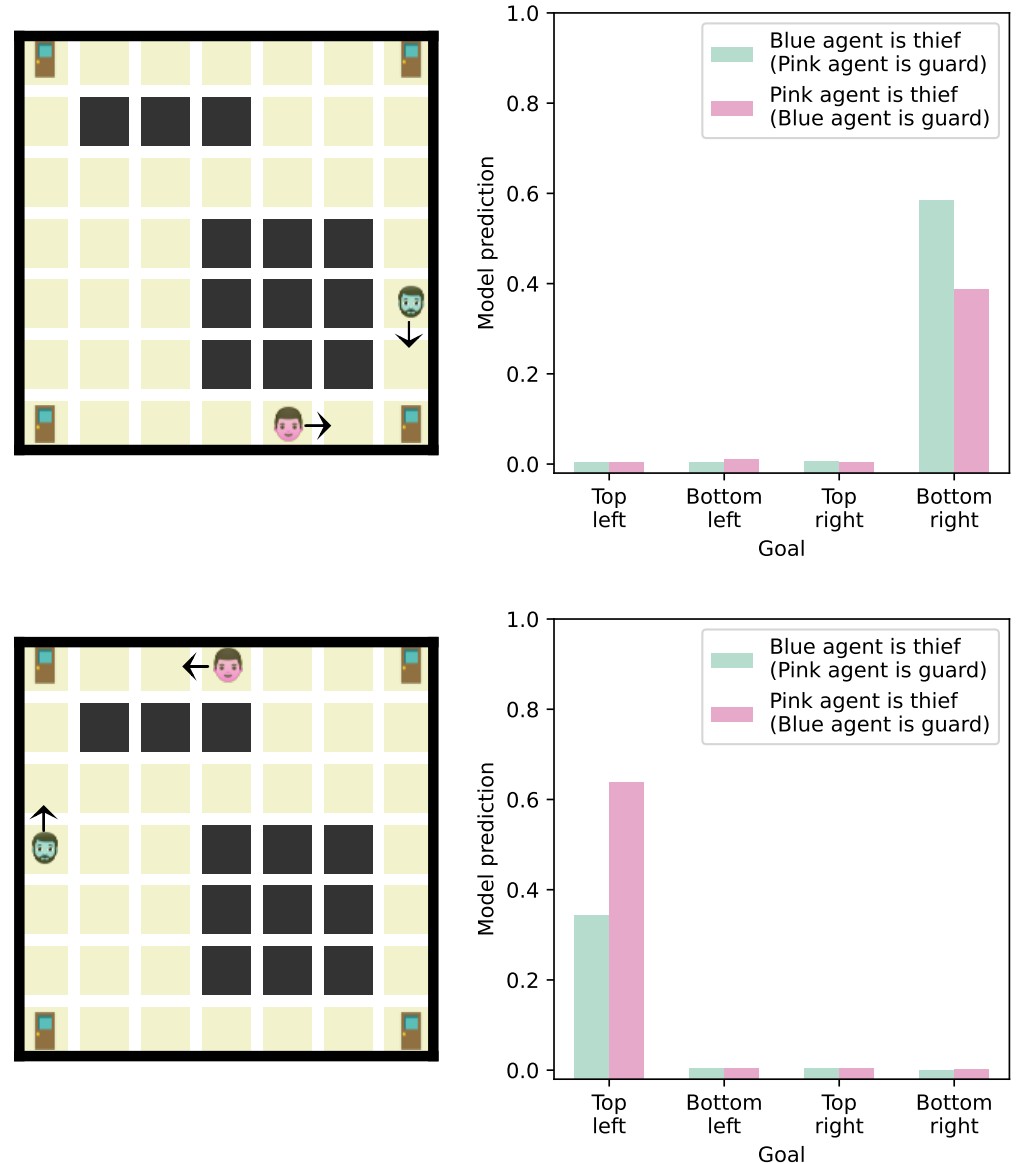

Figure 8: In these examples, it is unclear who the guard and thief are—however, it is clear where the exit is. **The model reproduces this uncertainty as desired.** See Appendix C.2.

The next two examples (Figure 8) are ambiguous cases: the two agents are in symmetric positions, so it is unclear who is who. Here, the model can determine with high confidence where the exit is, but remains uncertain about who is the thief and who is the guard.

Finally, in the last example (Figure 9), it is unclear whether a blue guard is blocking a pink thief from heading to the top-right corner, or whether a pink guard is blocking the blue thief from heading the the bottom-right corner. Indeed, the model reproduces this ambiguity.

## C.3 Cart-pole (continuous state space with physical dynamics)

The cart-pole domain is a classic problem in reinforcement learning and optimal control. The goal is to balance a pole in an upright position, by moving the cart left or right. The state space of this

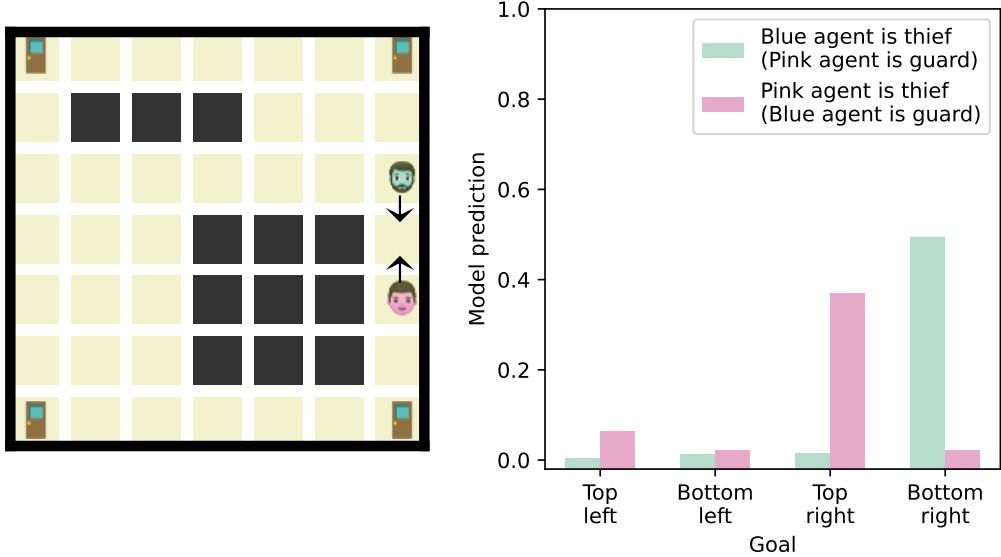

Figure 9: In this example, it is unclear whether a blue guard is blocking a pink thief from heading to the top-right corner, or whether a pink guard is blocking the blue thief from heading the the bottom-right corner. **The model reproduces this joint uncertainty as desired.** See Appendix C.2.

domain consists of four continuous numbers: the horizontal position of the cart and its velocity, and the angle of the pole along with its angular velocity. **The inference tasks are to look at a snapshot image—which only shows the cart position and the pole angle—and determine the velocity of the cart and the angular velocity of the pole.** Note that rejection sampling cannot solve this task because the probability of a randomly-sampled trace passing through the observed state is zero.

We use an off-the-shelf pre-trained Proximal Policy Optimization (PPO) controller [Schulman et al., 2017] from stable-baselines3 [Raffin et al., 2019] to compute a probability distribution over actions. Inference in this domain is complicated by the fact that computing backward dynamics in physical simulation is challenging and often ill-posed. While previous work has proposed analytic approaches [Twigg and James, 2008], we instead train a neural network to approximate the reverse physical dynamics. We place a unit Gaussian prior over the velocities, and use a von Mises distribution as a prior over the initial pole angle. We infer the velocities of the system by sampling candidate pairs of cart and pole velocities (stratified in an $11 \times 11$ grid) and computing likelihoods using our algorithm.

The inferred posteriors are intuitive and track the relative stability of the position in each snapshot (Figure C.3). For example, in part (a), the pole has almost completely fallen over, and so our method infers that the pole has a large negative angular velocity, and is falling fast towards the ground. At the same time, it infers that the cart is moving fast to the left, because the controller is likely attempting to re-balance. In comparison, for part (f), the pole is nearly upright, so the model predicts that the pole is not rotating, and that the cart might be moving left or right to keep the pole balanced.

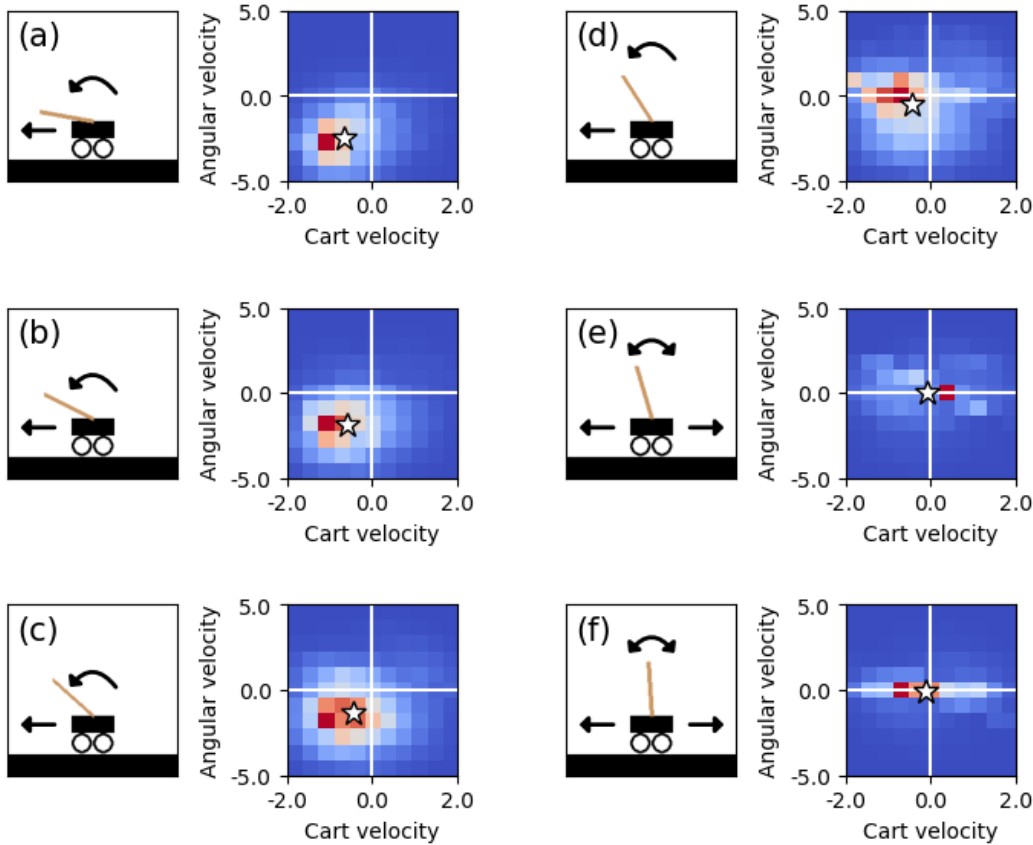

Figure 10: In each pair, the left image shows the cart-pole snapshot given to the algorithm, and the overlaid arrows summarize the model's predictions about how the system might evolve. The right heatmap shows our model's full joint distribution of inferred cart velocity (positive means moving to the right) and pole angular velocity (positive means clockwise), and the white stars mark posterior expectations. **When the pole is near-horizontal, our algorithm infers that the pole is falling, and the cart is moving left to re-balance. When the pole is near-vertical, the algorithm infers that the pole is stationary, and the cart is making minor adjustments to keep the pole balanced.**

