# OpenReview forum: "Inferring the Future by Imagining the Past"
_NeurIPS.cc/2023/Conference — NeurIPS 2023 spotlight_

### Official Review · Reviewer_PVLo · 2023-07-06

**Soundness:** 3 good
**Presentation:** 2 fair
**Contribution:** 2 fair
**Rating:** 4
**Confidence:** 3

**Summary:**

This paper presents an efficient Monte-carlo algorithm to infer goals from single snapshots. The problem the authors consider is the same as in work by Lopez-Brau (2020, 2022) but previous solutions are slow as they apply rejection sampling. In this work, the authors use the insight that one can sample a valid path by sampling the part of the path "before" the current snapshot and "after" the snapshot independently, decresing the computational complexity by a large amount. Further, A-star search is used to find likley paths. The authors validate their algorithm on different grid worlds and provide results of a human study indicating that inferences of their algorithm coincide with human judgements.

**Strengths:**

The problem of inferring goals from a single snapshot is interesting, as a major limitation of most IRL methods is that they only work in a certain context (state/actions of the full trajectory).

The paper is clearly written and easy to understand.

The presented method is much more efficient than previous methods in terms of scalability.

The plausability of the results found by the algorithm is supported by human experiments.

**Weaknesses:**

Quantities are not clearly defined. Are states/actions/goals continuous or discrete? In the formulas, one integrates over states and sums over actions. It might be very helpful for the reader to formally define the MDP and its sets. Further, what are "paths" formally, what does the indexing operator "[0]" mean for paths?

While the main idea of the algorithm to sample the path "before" and "after" is very simple, the overall algorithm seems quite hacky to me: The basic idea is combined with other tweaks such as roulette termination and A-star search to avoid unlikely paths. With these, the considered tasks can be efficiently solved but there is no particularly nice linking theory.

In my opinion, the contribution is not strong enough for a NeurIPS submission as the main (conceptual) difference to previously published algorithms is not that large. The essential difference is that parts of the path before and after are sampled, together with other tweaks.

I would find it helpful if the algorithm clearly indicated which quanitity it computes (p(x | g)), this is not direcly visible.

The plots in Table 1 do not have a colorbar, therefore the precise meaning and interval of the color's values is not clear to me.

**Questions:**

The regarded grid worlds all have a discrete state/action space. Is the algorithm applicable to the continuous domain?

When introducing the algorithm, you write that the situation of the agent you consider is an MDP but you would relax this assumption later in the paper. Where do you do that?

**Limitations:**

Limitations are well discussed. To my understanding, only discrete state/action/goal spaces are considered. This point could be added.

---

> ### Author Rebuttal · Authors · 2023-08-09
>
> Thank you for your thoughtful comments about our paper. We address your major concern about the lack of a linking theory in the **common response**. Here, we address your additional questions:
>
> **Is our method applicable to continuous domains?**
> Yes, it is: the cart-pole example in the supplement shows our method in a continuous state space. When revising, we will mention this in the main text. We will also update the paper to acknowledge that we do not address continuous _action_ spaces.
>
> **Where do we relax the MDP assumption?**
> Thank you for raising this — we will update the paper to clarify that we relax the MDP assumption in Sec 2.3. (Please see the common response regarding A-star for more information on why we do so, and how we will further revise that section based on the reviewers' comments.)
>
> **Improving presentation and definition of key quantities**
> Thank you for the suggestions for improving the paper's presentation. We will update the paper to:
> 1. introduce notation that defines the state/action spaces of MDPs,
> 2. formalize "paths" as finite ordered sequences of states, which are indexed and sliced the same as lists,
> 3. clarify that the output of Algorithms 1 and 2 is an estimate of $p(x \mid g)$, and
> 4. include colorbars in Tables 1 and 2.

---

> > ### Comment · Reviewer_PVLo · 2023-08-20
> >
> > Thank you for answering my questions. I still believe that the contribution is quite incremental therefore I keep my original score.

---

### Official Review · Reviewer_cLNU · 2023-07-06

**Soundness:** 3 good
**Presentation:** 4 excellent
**Contribution:** 3 good
**Rating:** 8
**Confidence:** 3

**Summary:**

This papers proposes a new approach to the inference problem of inferring an agent's goal state from information about a single state. This approach is based on the bidirectional Monte Carlo sampling of trajectory sequences, both from the agent's given state $x$ to the goal state $g$, and from an initial state $s$ to $x$. The authors show near convergent inference of the approach with 10 samples on two gridworld based environments, and a word game kind of environment when evaluated against human judgement.

**Strengths:**

- The evaluations show a strong "near" convergence result of only 10 samples, even for multi-stage planning. This indicates their sampling strategy is practically sample-efficient, and a huge improvement over prior approaches requiring more than 300,000 samples (reportedly).
- The connection drawn from path tracing in computer graphics is a fresh insight, which the authors show applies well to the goal/previous states' inference problem, and can potentially have broader scope applications such a for spatial navigation tasks etc.
- The use of human judgement (using 200 participants) for evaluation is also a very reasonable choice instead of hand-engineered reward functions designed by a single person/a few people.
- The authors also show efficient sampling in a variety of other domains (in the appendix) which indicates it is more broadly applicable.

**Weaknesses:**

- _Experiments_: Something I might myself be unclear about, so let me know if that is the case-- there are no evaluations comparing the proposed model in this paper with the model shown by [Lopez-Brau et al 2022](https://psyarxiv.com/4zu9n), is this only because of sample-efficiency? It would be interesting to see how "nearly" convergent their proposed approach is in the cases where it can be applied.

- _Theoretical analysis_: What is the complexity of the sampling with the size of the state space, the number of accessible states, the number of starting/goal states etc-- is it possible to perform some kind of theoretical analysis of this?

I'll be very happy to increase my score on the addressal of these concerns/doubts.

**Questions:**

- Have you checked how the sampling scales with an increasing number of possible starting states, and an increasing number of possible goal states? It could potentially be interesting to see.
- Why is incremental A-star search used for planning on-line?
- I am wondering if its possible to emphasize more on the scope of the applications of the proposed sampling approach-- what other domains can it be used in, apart from tasks that look like spatial navigation? The word blocks is one good example, also the cartpole experiment. There are intuitive physics based experiments, or reasoning based tasks which involve inferring over a sequence of images say, can this approach be applied there?

**Limitations:**

The authors adequately address the limitations of their approach, especially when it comes to sampling over goals, or scaling up to a larger goal space. There is just one minor point here-- the approach is based on a combination of well-founded insights and ideas, which when put together, work well, but there seems to be a lot of hand-engineering/domain-specific engineering in assembling these together, which I think should be acknowledged in the paper.

---

> ### Author Rebuttal · Authors · 2023-08-09
>
> Thank you for your thoughtful feedback on our paper. We address your questions below, and in the **common response**.
>
> **Did you compare to Lopez-Brau et al?**
> Yes, we indeed show these comparisons in Tables 1/2/3, under the heading of "Rejection." We apologize for the confusion and will revise the heading to make clear that that column refers to Lopez-Brau et al's method.
>
> **Theoretical analysis of scaling**
> We agree it would be good to address this, and we will update the paper with a discussion.
>
> The time complexity of taking a _single sample_ is straightforward: it scales linearly with the Russian Roulette termination depth and the number of possible goals, but is constant in the number of possible start states (because we sample backwards in time!). The plots in the **attached PDF file** establish this empirically using the grid-world domain; we will include such figures in the revision of our paper.
>
> (Runtime also scales linearly in the number of accessible "neighbor" states, because we softmax over possible actions. But in practice this number is fixed by the domain, and typically a small constant that contributes negligible overhead. For example, in our grid-worlds the number of accessible states is typically 4 for north/east/south/west.)
>
> Finally, another important factor is the _number of samples_ needed, which depends on how much variance is acceptable. Standard theoretical bounds for Monte Carlo integration hold here: variance reduces linearly with the number of samples, or even better via stratification [1, 2]. Such algorithms are well-studied in graphics [3, 4], and those tighter bounds may also be applicable to our setting.
>
> 1. Hickernell. Koksma‐Hlawka Inequality. 2014.
> 2. Bakhvalov. On the approximate calculation of multiple integrals. 2015.
> 3. Singh et al. Analysis of sample correlations for Monte Carlo rendering. 2019.
> 4. Subr et al. Fourier analysis of numerical integration in Monte Carlo rendering: theory and practice. 2016.
>
> **More non-spatial applications**
> We are glad the reviewers appreciated our non-spatial examples like blocks and cart-pole, and we are happy to expand our paper with a discussion of more such applications. As suggested, there is indeed scope for applications in intuitive physics and reasoning, such as:
> + Determining the viscosity of a fluid from a static image of it being splashed
> + Interpreting comic books or step-by-step instruction manuals, by inferring what happens between static panels
>
> Additional domains where our method could be useful include:
> + Robotics: walking into a kitchen and immediately recognizing what the chef is cooking, in order to help accordingly
> + Vision: understanding dynamic action in static snapshots of sports games
> + Forensics: inferring/reconstructing what occurred at a crime scene based on observable evidence
>
> **Acknowledging engineering work**
> Thank you for raising this - we will revise the paper accordingly. Please see the end of the **common response** for our full remarks on this point.

---

> > ### Comment · Reviewer_cLNU · 2023-08-19
> > **Response to rebuttal**
> >
> > Thank you to the authors for answering my questions and for the detailed clarifications!
> >
> > - sample complexity analysis: I see, makes sense. Thanks for attaching the plots, nevertheless.
> > - revision of writing and scope of work: The authors' response of revising the paper with respect to their contributions and motivation for using incremental A* seems reasonable to me.
> >
> > Thus, I increase my score to 8 for a Strong Accept.

---

### Official Review · Reviewer_yiSb · 2023-07-11

**Soundness:** 3 good
**Presentation:** 4 excellent
**Contribution:** 4 excellent
**Rating:** 8
**Confidence:** 3

**Summary:**

This paper presents an algorithm for efficiently inferring the goal of an RL agent from just observing its current (single) state $x$. The method  improves substantially upon rejection sampling based prior work by 1) only sampling paths through $x$ by separating the path into past and future 2) sampling the past path backwards in time starting from $x$ and 3) using incremental A-star search for planning.
The efficacy of the approach is demonstrated in two gridworld domains and a word-blocks game (as well as three more in the supplementary). The posterior estimates over the goal of the agent qualitatively match human judgements, and converges several orders of magnitude faster than vanilla rejection sampling.

**Strengths:**

* The studied problem is interesting and potentially impactful for multi-agent RL or human-agent interactions.
* The algorithmic modifications are well motivated and the efficiency improvement above simple rejection sampling is impressive.
* The paper validated the results against human judgements and found a high correlation
* The presentation is unusually clear, and a fun read.


**Weaknesses:**

The quantitative evaluation in Table 3 assumes the ground-truth to be a converged estimate of the presented method. This leaves open the possibility that the TV is better than the rejection sampling baseline because the presented method is biased in some way. The concern is partially addressed with the numerical validation check in the supplementary section B, which in my opinion should at least be mentioned in the main paper. Another way to strengthen the results in Tab 3 would be to also compare to a "ground-truth" estimate from (very?) many vanilla rejection samples. If the presented method is still better at approximating this ground-truth with few samples than rejection sampling with few samples, that would be strong evidence.


**Questions:**

* I am unclear about the role of A* in the proposed algorithm (sec 2.3). I assume that $p(x\rightarrow x') \propto \sum_a \exp (\beta(C(x \rightarrow g) - C(x' \rightarrow g)) $ takes the role of the A* heuristic. But in A* this heuristic is used to decide which node to further expand  next, and I cannot find the equivalent of this search process in Algorithm 2. Could you please clarify the correspondence of A* with the proposed algorithm.
* I am unclear about the shaded cells in in Table 1. The description states that "Shaded cells were excluded from the analysis because it would be irrational for the agent to be there for any goal".  I assume "shaded cells" refers to the gray cells in the rows corresponding to the keys world. But the shading of cells doesn't make sense to me. In row 3 it seems to correspond to inaccessible areas (which make sense to exclude). In row 4 the bottom third is shaded except for a single spot next to the green key. Why would that spot be less irrational than the surrounding ones? In row 5 the leftmost column is shaded which also seems odd. Is it impossible for the agent to start there? Also why are no cells shaded in row 1? Please clarify.


**Limitations:**

The authors discuss an important limitation of their work regarding the sampling over goals.
Another limitation that may be worth pointing out is that the separation of paths into independent paths for past and future depends on the environment to be Markovian (this *is* mentioned in the paper). More specifically that the current states $x$ contains all information connecting the past and the future. For more realistic environments this would for example have to include quantities like velocity or any relevant inner state of the agent in addition to the current position. Relaxing this assumption would require accounting for latent states and thus be equivalent to observing a distributions over current states $x$ instead of a concrete state.

---

> ### Author Rebuttal · Authors · 2023-08-09
>
> Thank you for your careful feedback on our work. Please see the **common response** for our remarks on the role of A-star. We respond to the rest of your questions below.
>
> **Table 3 ground truth**
> Thank you for raising this concern, which we will address as follows:
> 1. Mention the correctness checks in the main body of the paper, as suggested.
> 2. Expand Table 3 with two new columns showing analogous TVs compared to converged posterior estimates from many rejection samples, as suggested. For example, for the start-anywhere grid-world, the numbers using 10,000 rejection samples are below (essentially the same, as expected):
> | Method | Ground truth | Total variation |
> |---|---|---|
> | Rejection | Ours, 1k samples | 0.159 (from paper) |
> | Rejection | Rejection, 10k samples | 0.153 **(new)** |
> | Ours | Ours, 1k samples       | 0.0538 (from paper) |
> | Ours | Rejection, 10k samples      | 0.0836 **(new)** |
> 3. As an additional assurance, we can show scatterplots analogous to Fig 4 for our main benchmarks, to emphasize unbiasedness. This is essentially replotting the data in Tables 1/2. As an example, the scatterplot for grid-world is in the **attached PDF file** — the two candidate ground-truth posteriors match nearly perfectly.
>
> **Shaded cells in doors-keys-gems**
> We appreciate the detailed reading of our figures. Yes, the caption refers to the grayed cells in the keys domain, and the shaded cells in Row 3 are indeed shaded because they are inaccessible (we will make this clear when revising).
>
> As for the other rows, we will update the paper to explain our reasoning:
> + The shaded cells in Row 4 are excluded because if the agent already picked up the pink key earlier, there is no reason to _then_ move far _away_ from the doors. Similarly, in Row 5, there is no reason to move that far left after picking up the green key. We found that querying repeatedly at those extreme locations confuses human participants about the task setup. So, we excluded most such locations from our experiments.
> + The single cell above the green key in Row 4 is an exception we wished to test. You might think from that position that the agent is trying to collect _both_ keys, leading to uncertainty about the goal. But people (and the model) agree that the agent still wants the red gem.
>
> We will additionally clarify that the agent always starts empty-handed, which may be one source of confusion here.
>
> **Accounting for latent state information**
> Thank you for raising this limitation. We believe the cart-pole example from the supplement begins to address this: we make inferences when only positions, not velocities, are visible. It works exactly as you suggest, i.e. by observing a distribution over current states $x$. We will update the paper to explain this in more detail.

---

> > ### Comment · Reviewer_yiSb · 2023-08-18
> > **Response to rebuttal**
> >
> > I thank the authors for the clarifications, and for providing additional evidence for the unbiasedness of their method. Their response addresses my questions and concerns and would like to reaffirm my recommendation for a strong accept.

---

### Official Review · Reviewer_iHfX · 2023-07-26

**Soundness:** 3 good
**Presentation:** 2 fair
**Contribution:** 2 fair
**Rating:** 5
**Confidence:** 2

**Summary:**

This paper deals with the problem of inferring the goal state $g$ of an agent given only a single state $x$ in a trajectory, i.e., inferring $p(g|x)$.
For this, the authors claim that we need to integrate all possible initial states, and thus, sampling past trajectories is necessary for Monte Carlo estimation.
They propose several techniques to improve the sampling efficiency, and the main technique is to sample the past trajectory in reverse, starting from the given state $x$.

**Strengths:**

The key intuition and motivation are clearly described.

**Weaknesses:**

### The key assumptions are not clearly stated

I think the main text does not explain the key assumptions, such as
- What is given? A fully-trained agent? A fully transparent environment with perfectly known dynamics?
- How can we perform the past and future sampling?

### Soundness of the main argument

Most importantly, I am not convinced that explicitly sampling the past trajectory is necessary to infer $p(g|x)$.
Since $x$ contains all the information that we have, sampling the past trajectory seems like an unnecessary complication.
I think we can train a predictor that produces $p(g|x)$ directly from a decent amount of offline trajectories.

Moreover, I believe there is a serious mistake in Eq.(2).
The left hand side of Eq.(2) is $p(x|g)$, but the right hand side assumes that every path passes through $x$ and does not consider the cases where $x$ is not included in the path.
As the authors noted in L81-82, most paths would not pass through $x$, so there should be a term that account for this fact.

### Limited scope of the experiments

The experiments are limited to simple toy problems like grid world.

**Questions:**

- Please state the key assumptions
- Can you prove that sampling the past trajectory is absolutely necessary to predict $p(g|x)$?

**Limitations:**

Yes

---

> ### Author Rebuttal · Authors · 2023-08-09
>
> Thank you for raising several important points about our work. We believe we can address all of your concerns — we respond to them at the beginning of the **common response**.

---

> > ### Comment · Reviewer_iHfX · 2023-08-16
> >
> > Thank you for the response.
> >
> > I admit that I was wrong about Eq.(2). Paths passing through $x$ is expressed in the subscripts of $\pi$.
> > I also did not fully comprehend the underlying assumptions and problem setting, but after reading the response and other reviews, I think now I understand more clearly.
> > I'm raising my score, but still have some concerns and suggestions.
> >
> > - In addition to the stated key assumptions, I think the proposed method requires at least one more important assumption: for importance sampling into the past (L119-121), $p(s | s', g)$ should be easily computable.
> > This would be trivial if there are only a few possible $s$ for an $s'$, as in the grid world, but it can be challenging in other environments.
> >
> > - I felt the overall writing is a bit flashy.
> > The Hemingway and cognitive science stories sound far-fetched, compared to the actual algorithms and experiments.
> > I think toning it down can improve the delivery of key messages.
> >
> > - I think this paper has some novelty in academic perspective, but not sure if it can have practical values.

---

> > > ### Author Response · Authors · 2023-08-20
> > >
> > > Thank you for your comments. We are glad the discussion helped clear up any confusion in the original paper, and we appreciate the new suggestions.
> > >
> > > - **Past sampling:** Thank you for raising this — yes, we will revise the paper to note that we assume access to reverse transition dynamics. We will also:
> > >   1. Explain how the number of past/future “neighbors” of each state affects the algorithm’s runtime in a given environment (see the discussion on scaling with cLNU).
> > >   2. Highlight cart-pole as a case where reverse dynamics cannot be computed analytically, and were instead approximated with a neural network.
> > > - **Practical value:** We will update the paper to clarify that our primary goal is to model human intelligence, not to enable any particular application:
> > >   1. We will note that we are concerned with sample-efficiency specifically to explain how humans make such rapid intuitive judgements.
> > >   2. We will better situate our work in the cognitive science literature (referenced in our common response above).
> > >   3. That said, even though it is not our primary focus in this paper, our method *could* enable potential AI applications in the future (we highlight some examples in our response to cLNU under “More non-spatial applications”). We will add a discussion to the future work section of the paper.
> > > - **Writing:** Thank you for the suggestion — we are happy to adjust/tone down the writing based on the reviewers’ feedback. In particular, we realize that the references to cognitive science may seem extraneous because the current abstract and introduction do not explain that our goal is specifically to model human intelligence. We will adjust the writing to clarify this. For example, here are some candidate edits to the abstract:
> > >   > … **In this paper, we seek to model how humans make such rapid and flexible inferences,** even in domains they have never seen before. Building on a long line of work in cognitive science, we offer a Monte Carlo algorithm **whose inferences correlate well with human responses** in a wide variety of domains — while only taking a **small, cognitively-plausible number of samples.** Our key technical insight is to draw an analogy between our problem and Monte Carlo path tracing in computer graphics, which allows us to borrow ideas from the rendering community and dramatically increase the algorithm’s sample-efficiency.
> > >
> > >   And a more representative tl;dr:
> > >
> > >   > We **model how humans** infer an agent's goal from a snapshot of its current state. We frame the problem as Monte Carlo path tracing, which allows us to apply ideas from computer graphics to design a **cognitively-plausible sample-efficient algorithm**.

---

### Author Rebuttal · Authors · 2023-08-09

We thank all the reviewers for their thoughtful feedback. We address some key concerns below, and the rest in individual responses.

**Is sampling really necessary? Why not fit a neural network? (iHfx)**
Thank you for raising this important point. We realize that our paper's motivation was not fully clear, and we will revise the paper to explain the broader context of our work.

If our goal was only to solve this inference task in a particular environment, we could indeed just fit a model to predict $p(g\mid x)$ from datasets of offline trajectories generated in that environment. But here, we are interested specifically in how humans flexibly make such inferences _without_ extensive pre-training on data -- and how AI systems could do likewise.  A long line of empirical work in cognitive science, particularly in Theory of Mind, shows that people (even infants and young children) make rapid, flexible, and robust judgments "out of the box": in novel domains they have never seen before, and without extensive pre-training on data [1-5]. This remarkable one-shot ability is what motivates our work, and our sampling-based algorithm specifically seeks to capture that ability.

1. Gergely, G., and Csibra, G. "Teleological reasoning in infancy: The naive theory of rational action." _Trends in Cognitive Sciences_ 7.7 (2003): 287-292.
2. Baker, Chris L., Rebecca Saxe, and Joshua B. Tenenbaum. "Action understanding as inverse planning." _Cognition_ 113.3 (2009): 329-349.
3. Baker, Chris L., et al. "Rational quantitative attribution of beliefs, desires and percepts in human mentalizing." _Nature Human Behaviour_ 1.4 (2017): 0064.
4. Jara-Ettinger, Julian, et al. "Children’s understanding of the costs and rewards underlying rational action." _Cognition_ 140 (2015): 14-23
5. Hamlin, J. Kiley, Karen Wynn, and Paul Bloom. "Social evaluation by preverbal infants." _Nature_ 450.7169 (2007): 557-559.

**Limited scope of experiments? (iHfx)**
As part of the revisions promised above, we will explain that we based our experiments on tasks studied by empirical cognitive science research. Our experiments are comparable in scale to contemporary related work, both in AI (Zhi-Xuan et al, NeurIPS 2020; Shah et al, ICLR 2019) and cognitive science (Lopez-Brau et al, CogSci 2020). Furthermore, while prior work typically evaluates on only 1-3 domains, we consider a total of 6 domains across a wide variety of conditions (partial observability, multi-agent, continuous physics, etc.).

**Possible mistake in Eqn 2? (iHfx)**
Thank you for the attention to detail! This is a subtle point, but we still believe there is no mistake. We will update the paper to clarify: paths that do not pass through $x$ contribute zero likelihood of being observed at $x$, so we only need to integrate over paths that _do_ pass through $x$ (the "other term" would be zero). More generally, we hope that the numerical check in Supplement B assuages correctness concerns.

**Stating key assumptions (iHfx)**
We will revise the paper to clarify our assumptions: we assume full knowledge of environmental dynamics, and access to a planner to model how an agent would act given goal $g$. Formally, this is captured by $P(s \rightarrow s^\prime \mid g)$, a key input to Algorithms 1 and 2. Finally, we assume the ability to enumerate the neighbors of the current state, which are used for past/future sampling.

**Role of incremental A-star? (yiSb, cLNU)**
We realize that our motivation for using A-star was not clearly explained in Sec 2.3. We will revise the paper to clarify that we use A-star planning _only as an optimization_ — we can obtain the same results using value iteration, as in prior work, or for that matter any other planning algorithm (e.g. our cart-pole example uses deep RL).

We will additionally update the paper to better explain _why_ we use A-star:
1. Prior work formulates problems as MDPs, modeling agents' action choices by softmax over $Q(x, a)$. The drawback is that the full MDP must be solved offline before inference, e.g. by value iteration or deep RL. This is (a) cognitively implausible, and (b) wasteful if only some states are queried at inference time. Additionally, not all tasks are well-modeled by MDPs (e.g. multi-stage long-horizon planning).
2. Our solution is that where possible, we instead formulate problems as classical planning domains (c.f. PDDL) and weight actions by softmax over how much closer an action brings the agent to the goal. That is, for moving onto state $x^\prime$, we use the cost difference $C(x \rightarrow g) - C(x^\prime \rightarrow g)$.
3. The upshot is that we can compute the cost $C$ on-line by A-star search, avoiding the need for precomputation and only exploring relevant states. An additional optimization is to cache/memoize A-star's intermediate computations to avoid duplicate work if calls to $C$ are repeated.

We are happy to explain further if this remains unclear.

**Is this a "hacky" approach with no linking theory? (PVLo, cLNU)**
We realize that our exposition confusingly mixes theory and engineering, and we thank the reviewers for raising this issue. We will revise the paper to explain that there is indeed a core linking theory at play: the theory of Monte Carlo path tracing.

Rather than a grab-bag of "tweaks," we see our paper as organized around one key idea: the analogy between light paths and agent paths. This idea is what allows us to (1) reframe our problem using the theory of Monte Carlo path tracing, and (2) import existing engineering techniques motivated by that theory (e.g. Russian Roulette, bidirectional tracing) and thus solve our problem efficiently.

In short, we agree with cLNU's assessment: our work combines well-founded insights with some engineering work. We will revise the paper to distinguish the theoretical insights and engineering contributions we offer.

---

### Decision · Program_Chairs · 2023-09-21

**Decision:**

Accept (spotlight)

**Comment:**

This work leverages modeling from graphics to devise a cognitively plausible sampling method for inferring future states based on the previous trajectory.  The work compares to relevant baselines (including simple approaches like rejection sampling).  The reviewers are positive on the work with their primary concerns centering on framing.  Most issues are clarified in the author responses (e.g. issues around A*, the focus on cognitive plausibility, and clearly differentiating the novel modeling from engineering).  Personally, earlier discussion of the other domains (e.g. continuous cartpole) and what those imply about future broader application domains would strengthen the paper for those skeptical of the grid-world setup.